# Parental Supervision: Predictive Variables of Positive Involvement in Cyberbullying Prevention

**DOI:** 10.3390/ijerph18041562

**Published:** 2021-02-07

**Authors:** Jose M. Martín-Criado, Jose A. Casas, Rosario Ortega-Ruiz

**Affiliations:** Department of Psychology, University of Cordoba, Avda. San Alberto Magno S/N, 14071 Córdoba, Spain; m42macrj@uco.es (J.M.M.-C.); jacasas@uco.es (J.A.C.)

**Keywords:** family, cyberbullying, parental supervision, responsibility, competence, awareness

## Abstract

From an increasingly early age, parents face the challenge of educating their sons and daughters to act in the world of offline and online relationships. If for professional educators it is not proving easy, the involvement and guidance of parents in their children′s use of the internet seems to be a complex and unexplored challenge. This work aims to analyse the variables that influence digital education and determine a predictive model of positive parental involvement. This study was done with a representative sample consisting of five hundred and ninety-six families (596), representing the parents of children from schools with similar socio-cultural indexes. To do this, and using self-report instruments convertible into independent scales, four predictor variables were analysed: (1) parental knowledge of cyberbullying; (2) perception of parental competence in this regard; (3) parental perception of online risks; and (4) the attribution of parental responsibility in digital education. A structural equations model (SEM) examined the predictive value of these variables with respect to positive parental involvement. The structural equations model confirmed direct and mediated relationships between the independent and mediating variables on the dependent variable: parental supervision. The results indicate that positive parental involvement can be predicted from higher scores in parental knowledge of cyberbullying, perception of parental competence, risk adjustment, and attribution of parental responsibility.

## 1. Introduction

Information and communication technologies (ICTs) have become an interactive context where leisure, entertainment, and learning take place, also within family life. ICTs bring many advantages, such as immediate access to a large amount of information or the ability to communicate instantly with anyone [1]. Adolescents born in the digital age have grown up enjoying these advantages and communicate indistinctly in physical or virtual space [2]. However, these technologies also carry risks and pose new challenges for both school education and family involvement and guidance. Cyberspace is a medium where adolescents interact, often away from family supervision and sometimes in an aggressive and immoral manner [3]. In the last decade, both families [4] and teachers [5] have reported difficulties in teaching about the prevention of online risks, such as grooming, sexting, internet addiction, or cyberbullying [6] —the latter being one of the online risks that families are most concerned about due to its prevalence rates and harmful consequences [7,8] cyberbullying has been defined as a particular form of aggression that occurs when an individual or group uses digital devices to harm a person intentionally and repeatedly, who finds it difficult to prevent this harassment from continuing [9].

Most of the research on cyberbullying has taken a bio-psycho-social model as a reference, based on the ecological framework of Bronfenbrenner [10], in which different levels and individual (intrapersonal and interpersonal) and contextual [11] protective and risk factors have been determined. The family context, within this system, has been evaluated as one of the most relevant levels of influence in the prevention of cyberbullying [12].

### 1.1. Parental Supervision and Cyberbullying

Studies on family education have pointed out the important role that mothers and fathers can play in protecting their children from being victims of cyberbullying [13] the line of research addressing the prevention of cyberbullying in the family context has mainly taken two perspectives. The first has followed the previous line of research that related classic parental styles [14] the involvement in bullying on the internet [15,16].

Studies related to parental styles and online risks have shown, on the one hand, that the greatest risk for children is found in parental styles that combine excessive control and little involvement. On the other hand, families that provide affection, trust, and facilitate a fluid communication are related to safer and more competent children, thus reducing the possibilities of being involved in risk situations [17]. These results are in line with parallel studies that point out that authoritarian and negligent parental styles are related to a greater probability of children assaulting, especially if the parents apply physical discipline [18] while victimization is related to the permissive parenting style [19].

However, in relation to this first line of research and its method of analysis, it has also been pointed out that comparing parental styles and cyberbullying could be problematic [20] once most of the scales used [21] measure parental styles from the perspective of the children, and young people with more problems feel more distant from their parents and perceive them as more punitive, regardless of how the parents behave. In addition, the scales used to measure parental styles focus on the relationship between parents and children in a broad sense, but they can vary their parental style depending on the perceived self-efficacy or the tasks [20].

The second line of research has focused on parental supervision, understood as the combination of rules, accompaniment, communication, and control in the use of the internet and its virtual environments [22]. In relation to this second line, it has been pointed out that less competent parents tend to be less involved in active mediation, regulate inconsistently, and use more restrictive than communicative techniques [23]. Parental supervision has been recognized primarily as a protective factor [24,25]. However, recent systematic reviews [13,24,26] have shown inconsistent results among the not-so-numerous studies analysing parental factors.

On the other hand, given that parents are the adults most likely to be available at home, research on cyberbullying has also included a focus on the parents′ knowledge, understanding, and perception of the competence to combat it [27]. In this sense, it has been pointed out that the age and socio-educational level of parents are factors related to the digital divide [28]. In turn, the inequality in access to the internet and ICT, in combination with their beliefs, are factors that influence this implication [29]. In addition, it has been pointed out that parental beliefs are influenced by social factors and that parents are guided by their general parental style to regulate their children′s behaviour, although this is only partially effective in understanding how families become involved in specific behaviours such as cyberbullying [20].

### 1.2. Predictive Variables of Parental Involvement

Social cognitive theory [30] showed that cognitive processes are the first mediators of behaviour. It is known that family beliefs play an important role in the education of children in general and in the management of skills to prevent cyberbullying [29]. Parental knowledge and awareness are the first step towards prevention and intervention for children who are cyberbullies or cybervictims, as well as for those who witness cyberbullying [31]. So far, there is limited research that has examined parents′ awareness of their children′s online behaviour, especially in relation to exposure to cyberbullying.

Previous studies that have analysed parental beliefs in relation to cyberbullying have highlighted the need to improve the awareness of digital education [32] and have indicated that this awareness is affected by the third-party effect on parents, according to which parents perceive greater negative effects on other children compared to their own, and act on those beliefs.

The few studies carried out, most of them qualitative [27] have pointed out that the probability of parents′ involvement in the education of their children′s online behaviours is related to the concern and knowledge of the risks that such behaviour may include, the positive relationships of parents with their children, and the competence to address online behaviours [11]. In addition, research has shown that an adjusted perception and correct awareness of cyberbullying and the main online risks, as well as a good communication and supervision of children’s online activity by their parents act as protective factors [33]. On the other hand, research has also revealed that families and teachers differ significantly in their perception of coping with online behaviour and it is necessary to deepen the awareness of co-responsibility to coordinate their actions and be more effective in supporting students and children [34]. Conversely, lower levels of parental awareness and communication about their children′s online behaviour are related to greater inaction or neglect [35].

Despite the progress in cyberbullying research in recent years, the beliefs and attitudes of parents have hardly been addressed in a quantitative way. In this line, it would be necessary to develop studies that allow us to know and predict the involvement of families in the prevention of cyberbullying. To cover this space, this work aims to (1) analyse the parental beliefs related to a positive involvement in the prevention of cyberbullying; and (2) determine a predictive model of positive parental involvement in parental supervision.

With respect to the first objective, using independent scales, four variables were analysed: (1) parental knowledge of cyberbullying; (2) perception of parental competence in this respect; (3) parental perception of online risk; and (4) attribution of parental responsibility in cyberbullying prevention.

The first two variables, parental knowledge of cyberbullying and perception of parental competence, are understood as the degree of awareness, knowledge and competence to act on the main risks arising from internet use, associated with the concept of self-efficacy [30]. The parental perception of online risks, understood as the adjusted perception about the probability that their children are involved in cyberbullying, grooming, or sexting, is associated with the concept of risk perception [36] Finally, the fourth variable, attribution of parental responsibility, is understood as the belief in whether it is the teachers or the families themselves in the family or school environment who should mainly assume the task of preventing cyberbullying; it is related to the concept of attribution of responsibility [37].

The second objective of this work will be to examine the predictive value of these variables with respect to a better parental supervision. Based on these objectives, five research hypotheses are raised: (1) There is a relationship between the four predictive variables and parental supervision; (2) Families have an unbalanced perception of the online risks to which their children are exposed; (3) Families have an unbalanced knowledge of cyberbullying and other online risks; (4) Parents mainly attribute the responsibility of educating digitally to the school context; and (5) Families do not perceive themselves as competent to prevent cyberbullying (see Figure 1).

## 2. Materials and Methods 

### 2.1. Participants

The sample was made up of 596 families of which 76.1% of the responses were from mothers and the rest from fathers. The target population was families with children between 6 and 16 years old. With respect to their children: 45% studied in Obligatory Secondary Education, 14% in High School or Higher Level, and 41% in Primary Education. The families belonged to 51 educational centres selected among the eight provinces of Andalusia, Spain, which have similar socio-cultural indexes (ISC). The ISC is a government indicator obtained from the data provided by the families. An ISC is calculated in all the educational centres of Andalusia (Spain) from the information of the families on demographic data such as income, number of siblings, level of studies of the parents, neighbourhood, etc. A total of 64 middle level educational centres were selected for the study. Finally, 51 of them participated in the study and 13 educational centres decided not to participate, or their surveys were excluded due to response bias.

### 2.2. Instruments

This study explores the predictive variables of greater parental involvement in cyberbullying prevention, through five independent scales with good psychometric properties, based on previous research related to teachers′ perception of cyberbullying [38].

Two methods were used to establish the content validity of the scales: first, the scales were developed based on an existing, previously contrasted instrument developed by Li (2009) “Survey on School Cyberbullying for Preservice Teachers”. Although the original instrument was designed to know teachers′ perceptions about cyberbullying, valid parallels were found for families′ perceptions. The responses for each perception item were indicated by means of a five-point Likert scale, with responses ranging from totally false to totally true.

The existing instrument provided a solid basis for the development of the scales in terms of their validity. Second, a group of experts (including educators and psychologists) reviewed the scales. Finally, confirmatory factor analyses of each of the scales were calculated considering the proposed adjustment rates for categorical variables [39].

### 2.3. Procedure 

Permission was requested and obtained from school management, student parent delegations, and the school board for their participation in the data collection. Once the schools agreed on their approval, the data were collected. The questionnaires were filled out online by the families and the administration process took approximately 30 min. Before starting, all were informed about the voluntary nature of participation, the anonymity and confidentiality of the data, and the importance of responding sincerely. After removing 53 questionnaires due to response bias or incompleteness, 596 responses from 8 provinces were analysed. Regarding the statistical analyses performed to evaluate the psychometric properties of the scales, first a confirmatory factor analysis was performed from which the appropriate measures for 5 dimensions were derived. 

Secondly, the descriptive statistics of the items that make up the questionnaire were obtained and a model of structural equations was calculated. The models were estimated by means of the Least Square Robust method, adapted to the categorical nature of the variables under study [40]. The adjustment of the models was tested with the following indexes: Satorra–Bentler scaled chi-square (χ2S—B) [41] the comparative fit index (CFI), and the non-normality fit index (NNFI) (≥0.90 is adequate; ≥0.95 is optimal); the approximation mean square error (RMSEA) and the residual mean square (SRMR) (≤0.08 is adequate; ≤0.05 is optimal) [39] the Aiken information criterion (SIC) was used to compare the obtained models, where the best model has the lowest value.

## 3. Results

The objectives of this work were to analyse the variables related to parental beliefs associated to a positive involvement in family supervision and to determine a predictive model of positive parental involvement in family supervision.

Scale 1, parental knowledge of cyberbullying and other risks, with an α = 0.73, is composed of 5 items that describe aspects related to knowledge of cyberbullying, e.g., “A single message or image distributed by social networks could be considered cyberbullying”. The higher score obtained on this scale means better parental knowledge of online risks and their characteristics. The Confirmatory Factor Analysis (CFA) index: χ^2^ S-B = 132.65; *p* = 0.00; RMSEA = 0.02; SRMR = 0.08; IFC = 0.91; and NNFI = 0.92. It should be noted that almost all families (93.5%) say they are concerned or very concerned about their children′s online relationships. However, little more than half of parents (53.5%) fully believe that a single message could be the origin of a cyberbullying situation (see Table 1).

The second scale, perception of parental competence, with α = 0.81, is composed of 5 items that refer to the perceived ability to prevent, detect, and identify cyberbullying and other online risks, e.g., “I know strategies and tools to prevent online risks”. The higher score obtained on this scale means better perception of parental competence in educating and preventing online risks. The Confirmatory Factor Analysis (CFA) index: χ^2^ S-B = 3.45; *p* = 0.16; RMSEA = 0.03; SRMR = 0.02; IFC = 0.99; and NNFI = 0.98. The results indicate a low perception of competence to educate online. Just 37.5% of parents indicate knowing strategies to prevent online risks as true or totally true (see Table 2).

The third scale, called online risk perception, describes the adjustment in parents′ risk perception about the possibility of their children getting involved in cyberbullying and other online risks. e.g., “My children may be affected by Grooming”. The higher score obtained on this scale means a greater perception of risk in virtual environments. The Confirmatory Factor Analysis (CFA) index: χ^2^ S-B = 0.68; *p* = 0.70; RMSEA = 0.01; SRMR = 0.01; IFC = 0.99; and NNFI = 0.99.

It highlights the fact that almost a fifth of parents (19.8%) declare as true or totally true that their children behave differently online and that they are worried about getting likes (see Table 3).

Scale 4, attribution of parental responsibility for digital education, with α = 0.95, is composed of 10 items that indicates the extent to which parents think that cyberbullying prevention is a task for teachers at the schools or for the families at home. e.g., “The school context is more appropriate than the family context for prevention and intervention”. The higher score obtained in this scale attributes responsibilities to the educational centres and not to the families themselves. The Confirmatory Factor Analysis (CFA) index: χ^2^ S-B = 189.34; *p* = 0.00; RMSEA = 0.08; SRMR = 0.08; IFC = 0.98; and NNFI = 0.98.

Most families point to the co-responsibility of parents and teachers, although they point out that the school context is more appropriate for educating relationships online (see Table 4).

The final proposal includes one more scale not related to the original Li instrument. Scale 5, family supervision, with α = 0.81, is composed of 5 items related to parental supervision; it refers to the accompaniment, communication, and/or control exercised by families over their children′s online relationships, e.g., “I pay attention to my son/daughter when they surf …”. The Confirmatory Factor Analysis (CFA) index: χ^2^ S-B = 35.95; *p* = 0.00; RMSEA = 0.08; SRMR = 0.08; IFC = 0.95; and NNFI = 0.94. The results indicate that a little more than half of the families (55.2%) indicate as true or totally true paying attention to their children when browsing and having conversations about their online relationships (see Table 5). 

Regarding the first objective, the results obtained confirm the four hypotheses that derive from it. A high percentage of parents need to know more about cyberbullying and adjust their perception of the risks to which their children are exposed online, as well as to increase their perception of parental competence and the assumption of responsibilities to prevent the risk.

The second objective of this work is to analyse the predictive value of the variables described above with respect to positive parental involvement. But first, it must be stated that the poly-correlationships (see Table 6) show significant correlations between all the variables under study.

The model has an optimal setting with some indexes (χ^2^ S-B = 43.50; *p* = 0.90; RMSEA = 0.01; SRMR = 0.01; CFI = 0.99; and NNFI = 0.99). In Figure 2, the data derived from the structural equation model indicate that the perception of parental competence (PC) variable establishes the most important direct relationship with the dependent variable PS (β = −0.31), indicating that a higher score on the parental competence scale, and therefore a higher attribution of parental competence, is related to a lower score on the PS scale, which implies better parental practice. However, this PC variable is directly influenced by PKC (β = 0.33), which points to the important relationship between this parental knowledge and perception of online relationships and their risks and perception of parental competence. In fact, the PKC variable significantly influences the rest of the model variables; β = 0.35 for the ORP variable and β = 0.37 for the AR variable, indicating a great relevance in the model and its influence on the rest of the variables. As for the ORP variable, there is a direct relationship with the PS (β = 0.28), where the greater perception of risk is surprisingly related to worse parental practices. Finally, the variable AR is related to worse parental practices when the attribution of responsibility is in the school context (see Figure 2).

## 4. Discussion

The main objectives of this research were to analyse parents′ knowledge and beliefs related to cyberbullying prevention and to determine a predictive model of positive parental involvement. The results of the present study, in line with previous investigations, suggest that parents have limited knowledge of cyberbullying and the techniques they can use to prevent it. Parents who are involved reported using different techniques to monitor and guide their children′s digital behaviour and online relationships, protecting them from cybervictimization. However, this still does not seem to be a widespread educational practice among parents today. In fact, a significant percentage of parents still report that their children often surf alone and do not control or set limits on their online activity.

Regarding parental awareness, the results confirm the four hypotheses raised. About half of the parents who participated in the study reported having limited knowledge of cyberbullying and its risks. Likewise, low scores were found regarding the level of perception of online risks. In addition, parents point to co-responsibility and mostly think that this task must be shared with schools. However, parents with less knowledge and competence of cyberbullying tend to attribute to the school context the responsibility for online supervision. Therefore, it seems that many parents today, after some years since the first studies on cyberbullying awareness [38,42] still need to improve their knowledge, perception of competence and risks, and attribution of responsibility to prevent cyberbullying.

With respect to the second objective, the results confirm the hypothesis about the relationship between the four established variables: parental knowledge of online relationships and their risks; perception of parental competence; parental perception of online risk; and the attribution of parental responsibility in digital education as part of parental supervision. The structural equation models confirmed the direct and mediated relationships between the independent variables—parental knowledge of cyberbullying, the mediating variables, perception of parental competence, online risk perception, and attribution of responsibility—on the dependent variable—implication in parental supervision.

Perceived parental competence is the most important direct relationship in the model and the most influential factor for involvement in parental supervision. Parental competence, in turn, is directly influenced by parental knowledge of cyberbullying. Considering these relationships, one possible explanation for this lack of involvement can be found at the heart of social cognitive theory: self-efficacy beliefs. According to Bandura′s social cognitive theory, thoughts affect human functioning and are the primary drivers of behaviour. Therefore, improving the knowledge and perception of self-efficacy improves task engagement. Our results, in line with Bandura′s theory, indicate that parental involvement in cyberbullying prevention is a first requirement for developing effective online parental supervision. More precisely, a greater involvement is mainly related to parents′ knowledge about cyberbullying. Although, the perception of competence and responsibility to prevent cyberbullying are also influential factors.

## 5. Conclusions

For a whole generation of parents, cyberbullying prevention is a new educational task to be developed at home. Today, given the current levels of connectivity, this task has taken on renewed urgency. This study is important to determine not only whether parents have a clear understanding of what cyberbullying is, but also their level of involvement in preventing victimization. In this sense, this study contributes to the understanding of the factors that influence parents′ awareness of cyberbullying. 

It must be understood, consequently, that a positive parental involvement can be predicted from high scores in knowledge and perception of competence; from a parental adjustment in the appreciation of online risks and from a suitable attribution of parental responsibility to assume guidance and control in this new field of digital education. On the contrary, the attribution of the responsibility exclusively to the school, together with the ignorance or alienation of the social–digital life of their children, seem to be indicators of risk. It follows that families need to deepen their knowledge about online relationships and the risks they imply, but also to assume their own responsibility to intervene in this area of education. These conclusions are in line with current research that points out the importance for parents to know strategies of positive and safe parental mediation in the digital education of their children [31].

The results could be of great interest for parental involvement in the prevention of cyberbullying, at a time when digital behaviour is changing the forms of social relationships. Their results make it possible to evaluate and predict the quality of parental intervention, in a field that has so far only been studied with qualitative reports. In this sense, it is a step forward in measuring parental involvement in the prevention of victimization. For intervention programs, this can be a useful tool as a baseline for expanding parental supervision, focusing on those parents who may need it.

This study has certain limitations that should be considered when interpreting the results and the scope of the discussion. Mainly, the fact that this is a cross-sectional study implies punctual information on fathers and mothers with children of different ages, so subsequent studies could be designed with spaced intervals, which would perhaps give the findings greater predictive power. In addition, this work was conducted with self-reports, which, despite being the most widely used instruments in this field, may be affected by response bias or social desirability. This social desirability may be greater in this study because it is related to their reputation as parents, so future studies should include multiple informants or qualitative studies. In addition, it would be better to use cross-cultural studies in different contexts, which would allow the cultural variable to be considered when interpreting the results. Longitudinal studies could also explore the sociodemographic factors, such as age or gender. Although this work has limitations, this research reveals the need for children to be safer in cyberspace, increasing parents′ awareness and identifying resources (or the need for resources) to help parents educate and protect their children [27]. For all families, but especially for those with deficits in the predictive variables, this study reveals the need to develop policies, actions, and programs that favour their development, to face the challenge of educating cybercitizens in the coming years with greater guarantees. 

## Figures and Tables

**Figure 1 ijerph-18-01562-f001:**
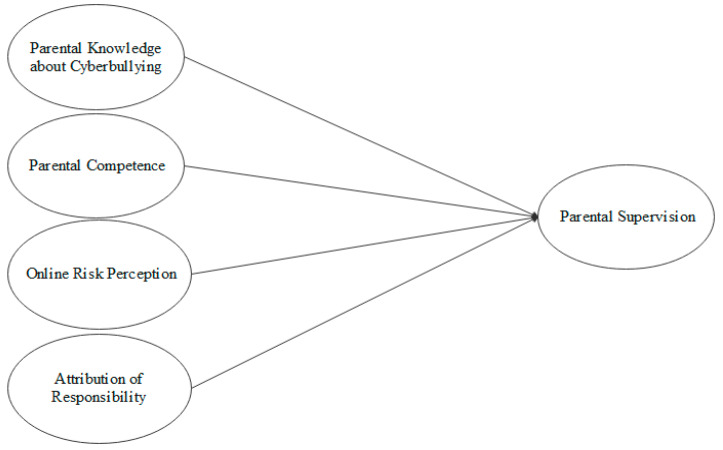
The hypotheses.

**Figure 2 ijerph-18-01562-f002:**
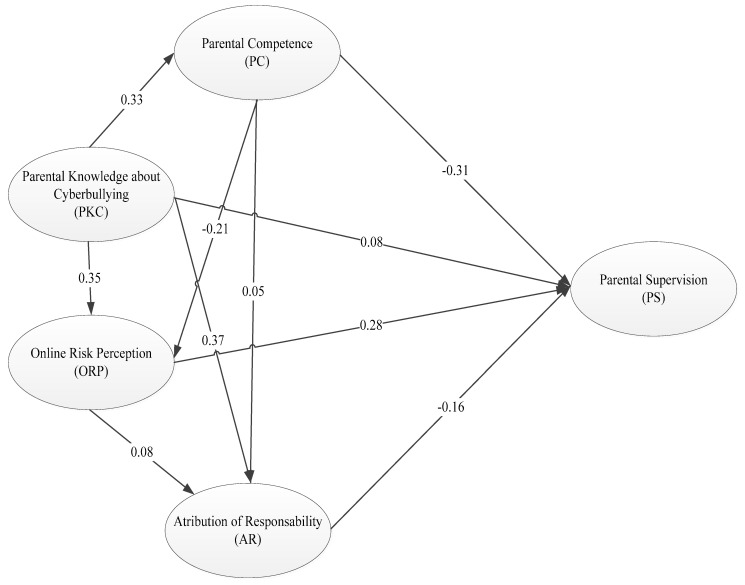
The system of structural equations.

**Table 1 ijerph-18-01562-t001:** Parental knowledge of cyberbullying.

TF = Totally False; F = False; NTNF = Neither True Nor False; T = True; TT= Totally True	TF	F	NTNF	V	TT	M	SD
I am concerned about cyberbullying	0.7%	0.3%	5.5%	14.6%	78.9%	4.72	0.63
A single message or image distributed by social networks could be considered cyberbullying	4%	5%	14.3%	23.2%	53.5%	4.18	1.09
Bullying and cyberbullying are equal phenomena, what changes is how they are attacked	0.8%	0.3%	4.4%	11.4%	83.1%	4.76	0.60
The main characteristics of cyberbullying are Intentionality, repetition, internet use	1%	0.7%	7.9%	24.5%	65.9%	4.54	0.74
Impersonating digital identities, hacking accounts or profiles in social networks are behaviours associated with cyberbullying	2%	1.5%	9.1%	20.6%	66.8%	4.49	0.87

**Table 2 ijerph-18-01562-t002:** Parental competence.

TF = Totally False; F = False; NTNF = Neither True Nor False; T = True; TT = Totally True	TF	F	NTNF	V	TT	M	SD
I know strategies and tools to prevent online risks	13.2%	19.8%	29.5%	20.1%	17.4%	3.09	1.27
I feel able to detect and identify cyberbullying	7.5%	17.1%	31.8%	26.3%	17.3%	3.29	1.15
I feel that I maintain a relationship of trust and fluid communication with my child about his/her online relationships and problems	4%	5.2%	22.1%	36.3%	32.3%	3.88	1.05
My child would come to me to tell me about a cyberbullying case if I was involved	1.7%	4.5%	22.9%	33.3%	37.5%	4.01	0.96
I think I am a good online role model for my child.	1.3%	2.5%	15.2%	37.2%	43.7	4.19	0.88

**Table 3 ijerph-18-01562-t003:** Online risk perception.

TF = Totally False; F = False; NTNF = Neither True Nor False; T = True; TT = Totally True	TF	F	NTNF	V	TT	M	SD
I upload photos of them regularly to RRSS, this does not put them at risk...	5.5%	16.8%	12.6%	16.8%	61.9%	4.28	1.12
My children may be affected by Grooming.	10.6%	7.7%	18%	15.8%	48%	3.84	1.37
My children may be affected by Sexting.	12.9%	8.4%	16.3%	17.4%	45%	3.74	1.42
My children behave more freely and uninhibitedly online. I think they are too concerned about “liking online”.	33.1%	20.1%	27%	11.2%	8.6%	2.42	1.28

**Table 4 ijerph-18-01562-t004:** Assumption of parental responsibility.

TF = Totally False; F = False; NTNF = Neither True Nor False; T = True; TT= Totally True	TF	F	NTNF	V	TT	M	SD
Cyberbullying is a concern in the educational community	0.7%	2.3%	11.4%	25.3%	60.2%	4.42	0.83
My child′s school should have concrete guidelines that can prevent cyberbullying situations	1.5%	1.2%	6.9%	22%	68.2%	4.55	0.78
Teachers, as responsible for this issue, should plan classroom activities to work on this problem.	1%	0.7%	6%	18.8%	73.5%	4.64	0.70
The parent’s association and/or the parent delegates should encourage activities to address cyberbullying	0.8%	0.8%	6.4%	22.8%	69.1%	4.59	0.70
The school context is more appropriate than the family context for prevention and intervention	0.8%	0.5%	7.4%	20.5%	70.8%	4.61	0.69
The orientation team/department should include tutoring sessions on cyberbullying	0.7%	0.2%	8.4%	22.3%	67.3%	4.71	0.59
My children′s centre should connect with resources outside the centre to address the problems of cyberbullying.	1.2%	0.8%	8.4%	22.3%	67.3%	4.54	0.78
I think it would be necessary to give more advice to students about cyberbullying	1%	0.2%	2.9%	13.8%	82.2%	4.77	0.58
I think it would be necessary to give more advice to families about cyberbullying	0.7%	0%	2.3%	12.9%	84.1%	4.81	0.51
I think it would be necessary to give more advice to teachers about cyberbullying	0.8%	0.5%	4.5%	14.4%	79.7%	4.73	0.63

**Table 5 ijerph-18-01562-t005:** Parental supervision.

TF = Totally False; F = False; NTNF = Neither True Nor False; T = True; TT = Totally True	TF	F	NTNF	T	TT	M	SD
My children are overusing social networks.	38.5%	20.3%	22.6%	9.5%	9%	2.30	1.31
I pay attention to my son/daughter when they surf, we have conversations about their online relationships. In addition, I control and set limits...	9.5%	14.1%	21.3%	25%	30.2%	3.52	1.30
My child surfs alone, I do not know who he/she is dealing with online. I do not control or set limits...	54.9%	16.8%	13.1%	7.9%	7.4%	1.95	1.27
My children behave more freely and uninhibitedly online. I think they are too concerned about getting likes.	33.1%	20.1%	27%	11.2%	8.6%	2.42	1.28
My child surfs alone, I do not know who he/she is dealing with online. However, I control and set limits.	32.5%	17.8%	22.8%	15.4%	11.6%	2.56	1.37
I pay attention to son/daughter when they surf, we have conversations about their online relationships. However, I do not control or limit	33.7%	21.1%	22.4%	13.6%	9.2%	2.43	1.31

**Table 6 ijerph-18-01562-t006:** Latent variable poly-correlationships.

	PKC	ORP	PC	PS	AR
PKC	1				
ORP	0.18	1			
PC	−0.40	−0.06	1		
PS	−0.23	0.42	0.32	1	
AR	−0.25	0.29	0.25	0.62	1

## Data Availability

The data presented in this study are available on request from the corresponding author. The data are not publicly available due to restrictions from the public administration.

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
