# Peer review of "Parental Supervision: Predictive Variables of Positive Involvement in Cyberbullying Prevention"

_ijerph, 2021, doi:10.3390/ijerph18041562_

Round 1

Reviewer 1 Report

Please see the attached notes with specific suggestions/recommendations/concerns.

Author Response

Reviewer 3

1) Add references to the digital divide and structural inequalities that shape ICT-related parenting practices for men and women.

Thank you for pointing out the convenience of adding references on the digital divide. The following references have been added in the paragraphs:

  1. “In this sense, it has been pointed out that the age and socio-educational level of parents are factors related to digital divide (Huang et al., 2018). In turn, the inequality in access to the internet and ICT in combination with their beliefs, are factors of influence for implication (Livingstone et al., 2014)
  2. It follows that the implication of fathers could be a fact to be considered for intervention with families. In this sense, research has also related a smaller digital divide with greater self-efficacy in digital parenting, which in turn implies greater prevention and protection of children from victimization (Huang et al., 2018).
  3. First, in line with previous studies (Bucea et al., 2020), the data indicate a possible continuity of gender stereotypes associated with the mother's role in digital education. Second, although normative gender pressure may make it difficult for mothers to disclose parenting difficulties (Markham et al., 2020), the overall results still reveal low competence.

2) Use of the term "Family education”:  I wonder if the use of the phrase "family education" "family education" is not confusing when compared to "school education"; the latter is more formal.

Following this recommendation, the use of the expression: "family education" has been replaced by the proposed expression: "parental guidance and control".

3) Motherhood. Significantly, most of the parents who responded were mothers. This relates to the limitations of the study that I will discuss below. It would be good to have the statistical trends disaggregated for any difference in parent responses (whether statistically significant or not)

Thank you very much for your suggestions. In relation to them the following modifications to the article have been introduced.

“It is noteworthy that almost 80% of the questionnaires sent to the families were answered by the mothers. This is important for at least two reasons. First, in line with previous studies (Bucea et al., 2020), the data indicate a possible continuity of gender stereotypes associated with the mother's role in digital education. Second, although normative gender pressure may make it difficult for mothers to disclose parenting difficulties (Markham et al., 2020), the overall results still reveal low competence”

Likewise, in a forthcoming study we plan to obtain data differentiated by age of the children and sex and age of the parents to deepen the analysis of these socio-demographic variables in relation to involvement in parental supervision.

4) About participating schools/schools and parent demographics. Participating schools/schools where parents came are described as possessing "similar sociocultural indices" (line 194). You should know more about this - what exactly are the indices? Are these high-, middle- or low-income neighborhoods? In general, the demographics of parents being recruited are important, as the results can be more easily applied to parents with lower general levels of education, SES, etc.

Thank you very much for your suggestions. In relation to them the following modifications to the article have been introduced. Changes have been introduced with word's change function: The ISC is calculated in all the educational centres of Andalusia (Spain) from the information of the families on demographic data such as: income, number of siblings, level of studies of the parents, neighbourhood, etc. 64 middle level educational centres were selected for the study. Finally, 51 of them participated in the study and 13 educational centres decided not to participate, or their surveys have been excluded due to response bias.”

5) Competence. I am not convinced that parental competence is being measured in this study, but rather parents' perceptions of competition - a subtle distinction perhaps, but that it can inflate the results. This can also be related to my point on the narratives of motherhood in society. The fact that mothers reveal their incompetence in raising their children calls into question the very strong social narratives about motherhood; admitting that she is a "bad mother," and the answers in this case can exaggerate "competition." This is also related to my interest in comparing the results between mothers and fathers.

Thank you for this contribution and this new point of view. In relation to them if you have replaced the name of the variable parental competence by “parental perception of competence”. In addition, considering the proposed idea, the following paragraph has been added:

“Although normative gender pressure may make it difficult for mothers to disclose parenting difficulties (Markham et al., 2020), the overall results still reveal low competence

6) About N. 596 questionnaires were received, but what was the total number initially distributed? In other words, what is the percentage of those who responded? Are there factors that help explain the lack of responses (e.g. lack of response from some schools, pandemic, etc.)

Thank you for pointing out the need to improve the information on N. More information on parent demographics, ISC and participation has been provided following this recommendation. Participants information has been modified adding more information about it. More precisely the following sentences have been included in the text: “64 middle level educational centres were selected for the study. Finally, 51 of them participated in the study and 13 educational centres decided not to participate, or their surveys have been excluded due to response bias.” and The ISC is calculated in all the educational centres of Andalusia (Spain) from the information of the families on demographic data such as: income, number of siblings, level of studies of the parents, neighbourhood, etc. 64 middle level educational centres were selected for the study. Finally, 51 of them participated in the study and 13 educational centres decided not to participate, or their surveys have been excluded due to response bias.”

 7) On Table 1 – a unique measure of cyberbullying. The definition offered in the literature review with respect to cyberbullying is accurate – it involves repeated cases of damage caused over time and through an imbalance of power between the victim and the victim(s). More should therefore be done in terms of the surprisingly high percentage of them (53.5%) who feel more strongly (i.e. 'totally true') than cyberbullying can constitute "a single message or image." Is this meant to demonstrate parents' ignorance of the "real" cyberbullying?

Thank you for the input on the difficulty of identifying parental knowledge of cyberbullying considering the conceptual approaches and nuances around its definition on the variable repetition. Certainly, the definition of cyberbullying has been labeled as fuzzy and "fluid"(Barlett, 2019), as conceptual approaches are expected to be open to changes that could be derived from technology and its use. In this sense, research has also shown that cyberbullying has specific features and nuances that derive from the use of electronic devices such as viralization (Hinduja, 2018, Casas et al., 2020). The inclusion of this item is based on the need for parents to know that a single image or video may be considered cyberbullying depending on its potential reach and viralization to improve prevention (although this sometimes escapes the control of parents and children themselves).

8) About the item ""My children behave more freely and uninhibitedly online. I think they are too worried about "like online." Doesn't this capture two separate things? Also, what "like online" means is ambiguous. In addition, parents (especially mothers) may well deny this item as fake given the effect of "pink colored glasses" who see their children through (i.e., refuse to see their own children as problems, in this case online).

Thank you for pointing out the need to improve the translation regarding this item. Certainly, the expression “like online” can be confusing, in fact the expression in its original Spanish version refers to getting likes. (it has been corrected in the tables) Also, considering your input, the rose-colored glass effect has been included in the text. For this, some references have been included related to the third person effect on parents and the need to take the results with caution due to the expected social desirability of the responses. Despite this, the results demonstrate a lack of parental competence.

9) Study limitations. Line 382 mentions the limitations of the study without detailing them.

Thank you for pointing out the need to improve the limitations of the study. Based on this, the limitations have been rewritten and changes have been made with word change control. The limitations are described as follow: “With respect to the limitations of this work, we must consider those of most studies that include self-reports, where social desirability and parental third-person perceptions may affect the results. This social desirability may be greater in this study because its related to its reputation as parents. For this reason, future longitudinal studies should deepen the relationships between parental knowledge and awareness, the strategies developed and the online performance of children. Longitudinal studies could also explore sociodemographic factors such as age or gender, as well as causal relationships that were not studied in the present study, due to its cross-sectional design. Although this work has limitations, its results are of great interest for parental involvement in cyberbullying prevention family educational action, in an era where digital behaviour is changing the forms of social relationship. Its results allow us to evaluate and predict the quality of parental intervention, in a field that until now has only been studied with qualitative reports.”

10) Paper reads well, in terms of writing, although several passages require some grammatical corrections or attention to syntax errors:

Line 30: Information and Communication Technologies (ICTs) have become an interactive scenario

Line 40: Educate Online Relationships and...

Line 47: But the psycho-evolutionary process is not simple but multidimensional

Lines 272, 273: The information, which the families themselves have provided, indicates

Line 288: The belief about who is the main responsible for digital education

Lines 356, 357: cybervictimization, but this seems to depend on parents, are perceived as responsible and digitally competent,

Line 370: high scores in the domain itself and competition

Thank you for pointing out the need to correct the sentences on the lines indicated. Based on this, the sentences have been rewritten and the changes have been made with word's change control.

11) Table 3 also includes several commas that replace decimal points - these must be reverted to decimal points (for example, 16.8% must be 16.8%).

Thank you for pointing out the need to correct the tables, all the decimal points have been revised.

Reviewer 2 Report

The introduction and problem are well presented, but it is very long (lines 0 to 187 : almost half of the paper). It should benefit to be shorten, focussing on the more important data. 

Materials and Methods

This section must be improved to describe more precisely the method. What about sampling ? How the 596 families have been selected among the eight provinces ? Randomly? From which population ? What is the response rate ? 

Line 189: report stats on mothers and fathers in results. Add other informations about the population in results. 

Lines 210 - 240 : This information should be presented in tables (1 to 5) in the results section, so the «procedure» would appear after line 209.

Results

This section must be improved to be publish. Information tend to let us think that statistical analysis is well done, but there is a major lack in the results presentation.

Line 268 : «many families (47,5%)» : where does this information come from ? We can't find it in table 1.

Tables 1 to 5 : Tables should presented the median (mode) of each item, and the results of the overall index (mean, standard deviation, alpha, results of Confirmatory Factor analysis - CFA). 

For each table, in the text, results should described overall index results, not just parsimonious and random information on specific items. 

Line 274: Where does 37,5% come from ?

Line 279 : Where does 61,2% come from ?

In this form, conclusions are not in line with the results presented. (Ex: Lines 324 to 330 : links must be exposed between conclusions and results).

Line 337: PPED : this abbreviation does not appear in figure 2. Idem for PPED (line 339), CPROR (line 340), PFRO (line 343). 

Discussion and conclusions

Lines 353 to 360: Sentence must be revised.

Overall, this section must be more in line with the results. Limits must be described more precisely. 

Lines 374 to 376: how can we conclude that ? Links are not well explained.

Author Response

Reviewer 2

  1. a) The introduction and problem are well presented, but it is very long (lines 0 to 187: almost half the paper). It is beneficial to shorten, focusing on the most important data.

Thank you for pointing out the need to shorten the introduction. Based on this recommendation, some parts of the text have been eliminated. The changes have been noted using the word function to facilitate the revision.

  1. b) Materials and methods: This section should be improved to describe the method more accurately. What about sampling? How have the 596 families been selected from the eight provinces? What is the response rate?

The section methodology and participants has been modified to include this information. In relation to the selection, it was a non-random, incidental sampling for accessibility. It is included in the text. All changes have been introduced with change control to facilitate review.

  1. c) Line 189: Report statistics on mothers and fathers in the results. Add other information about the population in the results. Lines 210 - 240: This information should be presented in tables (1 to 5) in the results section, so the "procedure" would appear after line 209.

Thank you for this contribution, Lines 210 - 240 have been modified in the text. This information has been included in the results section behind each table, as recommended.

 Results: This section must be improved to be published. Information tends to make us think that statistical analysis is well done, but there is a large deficit in the way the results are presented

  1. d) Line 268 : "many families (47.5%)" : where does this information come from? We can't find it in Table 1.

The percentages and the information related to the tables have been better explained following the recommendations of the reviewer 3.

  1. e) Tables must present the median (mode) of each item, and the results of the global index (mean, standard deviation, alpha, confirmation factor analysis results - CFA) the cfa if you are adding the other.

Thanks for the correction. Two columns have been added to the tables with the m and SD measurements. Likewise, the inclusion of the rest of the alpha and CFI measures has been revised and included in the description of the scales.

  1. f) Line 274: Where does 37.5% come from? Line 279. Where does 61.2% come from? 274: The percentages and the information related to the tables have been better explained following the recommendations.

  1. g) The conclusions are not in line with the results presented. (Ex: Lines 324 to 330 : the links between the conclusions and the results should be exposed). Thank you for highlighting the need to improve this section. The conclusions have been revised to highlight their relationship with the results obtained in the report. Changes have been introduced with change control to facilitate review.

  1. h) Line 337: PPED : This abbreviation does not appear in Figure 2. Idem for PPED (line 339), CPROR (line 340), PFRO (line 343).

Thanks for the corrections about the abbreviations, the translation of these has been revised. The current corrected paragraph is as follows “The model has an optimal setting with some indexes (χ² S-B = 43.50; p = .90; RMSEA = .01; SRMR = .01; CFI = .99; NNFI = .99). In Figure 2, the data derived from the structural equation model indicate that the Parental Competence (PC) variable establishes the most important direct relationship with the dependent variable PS (β = -.31), indicating that a higher score on the parental competence scale, and therefore a higher attribution of parental competence, is related to a lower score on the PS scale, which implies better parental practice. But this PC variable is directly influenced by PKC (β = .33) which points to the important relationship between this parental knowledge and perception of online relationships and their risks and perception of parental competence. In fact, the PKC variable significantly influences the rest of the model variables (β = .35) to the ORP variable and (β = .37) to the AR variable, indicating a great relevance in the model and its influence on the rest of the variables. As for the ORP variable, there is a direct relationship with the PS (β = .28) where the greater perception of risk is surprisingly related to worse parental practices. Finally, the variable AR is related to worse parental practices when the attribution of responsibility is in the school context.”

  1. i) Discussion and Conclusions Lines 353 to 360: The paragraph should be revised. In general, this section should be more in line with the results. Limits should be described more accurately.

Thank you for highlighting the need to improve this section. The conclusions have been revised to highlight their relationship with the results obtained in the report. Changes have been introduced with change control to facilitate review.

Reviewer 3 Report

This paper produces important findings linking parental supervision with cyberbullying intervention and intervention. The literature review and theoretical framework incorporate relevant sources and justify the need for the present study aptly. The findings will be of particular interest to academics but also parents and parent-teacher associations, as well as educators. I read this paper overall positively, but do have some suggestions for minor revisions which will improve the impact of the paper. Centrally, a few more references in the literature review to digital divide, and structural inequalities shaping parenting practices related to ICTs, unpacking results with greater attention to participant demographics, and improving the discussion based on these changes. I will detail these in turn below.

1: use of the phrase ‘family education’

I wonder if 'family education' isn't confusing when compared with 'school education'; the latter being more formal. Parents are engaged in a range of practices, many informal and involve monitoring, that doesn't easily equate with 'education', and the surveys also capture aspects that extend beyond ‘parenting education’ per se. I would simply unpack what is meant by this with a different phrase – perhaps ‘parental involvement and guidance’ or something along those lines. I’m sure most readers will understand what is meant here, but ‘family education’, especially given home-based schooling during the pandemic, may remind some of home schooling practices, which is not what is being examined here.

2: motherhood

It’s significant that the majority of parents who responded were mothers. This ties into study limitations which I’ll discuss below. I anticipated seeing statistical trends unpacked for any differences in responses between fathers and mothers (whether statistically significant or not).

3: participating schools/educational centres and parental demographics

The participating centres/schools from which parents were drawn are described as having “similar socio-cultural indexes” (line 194). More should be made of this – what are the indexes exactly? Are these high, middle or low income neighbourhoods? More in general on the demographics of parents being recruited is important, given that the findings may apply more readily to parents with lower overall levels of education, SES, etc.

4: interpreting parental competence

I’m not convinced parental competence is being measured in this study, but rather parental perceptions of competence – a subtle distinction perhaps, but one which may inflate results. This may also link back to my point about motherhood narratives in society. For mothers to reveal incompetence in parenting challenges very strong societal narratives regarding motherhood; admitting to being a ‘bad mother’, and responses here may exaggerate ‘competence’. This is also related to my interest in comparing results between mothers and fathers.

5: response N.

596 questionnaires were received, but what were the total number initially distributed? What, in other words, is the percentage of those who responded? Are there any factors that help explain where responses were lacking (e.g., lack of response from some educational centres, the pandemic, etc.)?

6: table 1 – single measure of cyberbullying

The definition offered in the literature review regarding cyberbullying is accurate – it involves repeated instances of harm produced over time and through a power imbalance between victimizer(s) and victim(s). More should be made, therefore regarding the surprisingly high percentage of those (53.5%) who feel most strongly (i.e., ‘totally true’) that cyberbullying may constitute “a single message or image”. Is this meant to demonstrate parental ignorance regarding ‘real’ cyberbullying?

Relatedly, the literature review can highlight several recent critiques of the ‘discourse’ of cyberbullying. Researchers have been finding that young people can easily dismiss harmful actions online as ‘pranking’ or ‘joking’ etc. while (sincerely) believing that ‘cyberbullying’ is harmful. Parents likewise may not view the actions of their own children as ‘cyberbullying’; of course this can reinforce the views of young people that their actions are not producing harm.

7: table 3 – final item

I’m confused about the item ““My children behave more freely and uninhibitedly online. I think they are too concerned about "liking online".” Doesn’t this capture two separate things? Also what “liking online” means is ambiguous. Moreover, parents (mothers especially) may well disavow as false this item given the ‘rose coloured glasses’ effect they see their children through (i.e., refusing to see your own children as having troubles, in this case online).

8: age and grade

Some of the findings, especially those in table 5, offer some interesting generalities but the findings would be greatly enhanced by unpacking trends related to the age of the children being considered (age and grade). A child “surfing alone” in senior high school is likely to be far less of an issue than a child doing so in elementary school.

9: study limitations

Line 382 mentions study limitations without detailing what these are, except for a passing reference to social desirability mentioned at lines 392-393. However given my points above, more details regarding qualifications and limitations – which of course can be pitched as directions for future research – should be explicated here.

The paper reads well, in terms the writing, though several passages require some grammatical fixes or attention to errors in syntax:

Line 30:

 Information and communication technologies (ICT) have become an interactive scenario

Line 40:

educating online relationships and

Line 47:

But the psycho-evolutionary process is not simple but multidimensional

Lines 272, 273:

The information, that the own families have contributed, indicates

Line 288:

belief about who are the main responsible in digital education

Lines 356, 357:

cybervictimization, but this seems to depend on the parents, are perceived as responsible and digitally competent,

Line 370:

high scores in the own domain and competence

Table 3 also includes several commas which substitute for decimal points – these need to be reverted to decimal points (e.g., 16,8% should be 16.8%)

Best wishes to the authors for successful revisions for this interesting paper!

Author Response

(The authors gave the same response as above.)

Round 2

Reviewer 2 Report

Authors upgraded the paper correctly. Limits should still be better described.

Author Response

Thank you very much for your comments. The New version of manuscript change the limits of investigation.